# The Monocytic Cell Line THP-1 as a Validated and Robust Surrogate Model for Human Dendritic Cells

**DOI:** 10.3390/ijms24021452

**Published:** 2023-01-11

**Authors:** Johanna Maria Hölken, Nicole Teusch

**Affiliations:** Institute of Pharmaceutical Biology and Biotechnology, Heinrich-Heine University Düsseldorf, Universitätsstraße 1, 40225 Düsseldorf, Germany

**Keywords:** dendritic cells, THP-1, DC maturation, cytokine, sensitization, phagocytosis, nickel sulfate, NiSO_4_, h-CLAT, 1-chloro-2,4-dinitrobenzene, DNCB, interleukin-12, IL-12

## Abstract

We have implemented an improved, cost-effective, and highly reproducible protocol for a simple and rapid differentiation of the human leukemia monocytic cell line THP-1 into surrogates for immature dendritic cells (iDCs) or mature dendritic cells (mDCs). The successful differentiation of THP-1 cells into iDCs was determined by high numbers of cells expressing the DC activation markers CD54 (88%) and CD86 (61%), and the absence of the maturation marker CD83. The THP-1-derived mDCs are characterized by high numbers of cells expressing CD54 (99%), CD86 (73%), and the phagocytosis marker CD11b (49%) and, in contrast to THP-1-derived iDCs, CD83 (35%) and the migration marker CXCR4 (70%). Treatment of iDCs with sensitizers, such as NiSO_4_ and DNCB, led to high expression of CD54 (97%/98%; GMFI, 3.0/3.2-fold induction) and CD86 (64%/96%; GMFI, 4.3/3.2-fold induction) compared to undifferentiated sensitizer-treated THP-1 (CD54, 98%/98%; CD86, 55%/96%). Thus, our iDCs are highly suitable for toxicological studies identifying potential sensitizing or inflammatory compounds. Furthermore, the expression of CD11b, CD83, and CXCR4 on our iDC and mDC surrogates could allow studies investigating the molecular mechanisms of dendritic cell maturation, phagocytosis, migration, and their use as therapeutic targets in various disorders, such as sensitization, inflammation, and cancer.

## 1. Introduction

Dendritic cells (DC) are sentinel leukocytes mediating the innate and adaptive immune response in mammalian solid tissue. Dendritic cells play fundamental roles in infections [1,2], inflammation [3], skin sensitization [4], and allergy [5], as well as in cancer [6,7,8] and are, therefore, of great interest in research [9]. The DCs are a heterogenous population of cells, specialized in antigen presenting. Upon exposure to, for example, pro-inflammatory cytokines, such as tumor-necrosis factor alpha (TNF-α), bacterial agents, such as lipopolysaccharides (LPS), or chemically-derived haptens, such as from nickel sulfate (NiSO_4_), maturation and migration of DCs is initiated. The DCs initially transform into immature dendritic cells (iDCs) with high endocytic activity and low T-cell activation potential [10,11]. The first steps of phagocytosis and maturation are accompanied by upregulation of the major histocompatibility complex (MHC) class II, such as the Human Leukocyte Antigen–DR isotype (HLA-DR) [12,13]. The MHC II molecules are synthesized on the cytosolic surface of the endoplasmic reticulum and are chaperoned by the invariant chain (li) to the late endosomal compartment where they encounter and fuse with endosomes loaded with the exogenous protein of presenting, building lysosome-like antigen processing compartments [14,15,16]. In order to bind antigens, the li peptide is cleaved. After loading a peptide derived from the exogenous protein, the class II molecules are exported to the cell surface for recognition by CD4^+^ T cells [15]. In addition, internalized antigens can be loaded onto MHC I molecules for cross-presentation to CD8^+^ T cells [17,18]. Simultaneously, expression of adhesion molecules, such as clusters of differentiation (CD)54 and co-stimulatory molecules, such as CD80 and CD86, are upregulated and transported to the cell surface, inducing dendritic cell maturation [19,20]. Furthermore, DC maturation is accompanied by upregulation of migration markers, such as the C-X-C chemokine receptor type 4 (CXCR4) and C-C chemokine receptor type 7 (CCR7), resulting in antigen presentation to T cells in lymphoid tissues [21,22]. Overall, migration of DCs is a complex process which depends on chemokines, such as stromal cell-derived factor 1 (SDF-1), chemokine (C-C motif) ligand (CCL) 19, and CCL21 [23,24,25,26]. The SDF-1 is secreted by fibroblasts and endothelial cells in the dermis after antigen exposure, inducing chemotactic migration of CXCR4-expressing DCs to lymphatic vessels in the dermis [23,24]. Contrarily, CCL19 and CCL21 are secreted by lymph nodes, inducing CCR7 dependent migration of DCs and naïve T cells [25,26,27].

The activation of naïve CD4^+^ T cells is initiated by the interaction of T cell receptors (TCRs) with the antigen-loaded MHC II complexes [28,29]. However, to fully prime naïve CD4^+^ T cells, the simultaneous interaction with DC-expressing CD54, CD80, and CD86 is required [30,31]. In order to form a stable signaling structure between dendritic cells and naïve CD4^+^ T cells, the intracellular adhesion molecule (ICAM-1)/CD54 forms with its partners, leukocyte function-associated antigen 1 (LFA-1) alpha (CD11a) and beta-2 (CD18), a cell–cell adhesion, the so-called immunological synapse (IS) [32,33]. Subsequently, high expression of the surface molecules CD80 (B7-1) and CD86 (B7-2) on antigen-presenting cells (APCs) allows the co-stimulation of naïve CD4^+^ T cells via their CD28 and cytotoxic T-lymphocyte-associated protein 4 (CTLA-4)/(CD152) receptors [34,35]. Upon the cell–cell contact between dendritic cells and T cells, several signal cascades in both T cells and dendritic cells are initiated, depending on the stability and duration of the cell-cell contact, and the number of MHC complexes and co-stimulatory molecules enhancing the transfer [36]. In immature dendritic cells, CD83 is not expressed on the cell surface, but is stored in the Golgi complex and endocytic vesicles, and the receptor can be transported to the cell surface immediately upon maturation [37,38]. Notably, CD83 knockout studies revealed a severe reduction in CD4^+^ T cells, proving the essential role of CD83 for the development of CD4^+^ T cells [39,40]. By binding to the membrane-associated RING-CH8 (MARCH-8) ubiquitin ligase, which is responsible for the internalization of MHC II, CD83 stabilizes the MHC II surface expression [41]. In addition, transmembrane regulation of the MARCH-1 ubiquitin ligase promotes the upregulation of surface MHC-II and CD86 on activated DCs [42,43], ensuring the stimulation, proliferation, and maturation of naïve T cells into primed effector and memory T-lymphocytes in the draining lymph nodes [44].

In the past decades, the predictive identification of potentially sensitizing agents has been performed via the guinea pig Buehler test or the murine lymph node assay LLNA. However, due to the differences between human and guinea pig/murine skin physiology and immune biology many agents were classified as false positive or negative [45,46]. Hence, there was an urgent need for alternative robust human-derived models. In this context, various protocols generating dendritic cell surrogates derived from human donor-derived peripheral blood monocytes (PBMCs) to study skin sensitization and inflammation have been reported [47,48,49,50]. However, the isolation and differentiation of PBMCs comes with various technical and biological limitations, such as the amount, availability, and donor heterogeneity [51,52]. In 2006, the human cell line activation test (h-CLAT) was developed by Ashikaga et al. [53,54]. The h-CLAT addresses one of the key events of the skin sensitization, by measuring CD54 and CD86 as markers for dendritic cell activation on the monocytic cell line THP-1 [53], originally isolated from peripheral blood of an acute leukemia patient [55]. The method is designed to distinguish between sensitizing and non-sensitizing agents, where the chemicals 1-chloro-2,4-dinitrobenzene (DNCB) and nickel sulfate (NiSO_4_), both acting as strong sensitizers, are the positive controls. In order to be classified as a sensitizer, the relative fluorescence intensity (RFI) has to exceed a defined threshold, which is CD54 ≥ 200 or CD86 ≥ 150 in at least two out of three independent measurements [56,57]. Due to high intra- and inter-laboratory reproducibility (80%) [57,58], the h-CLAT was validated and authorized by the European Union Reference Laboratory on Alternatives to Animal Testing (EURL ECVAM) and by the Organization for Economic Co-Operation and Development (OECD) for the toxicological assessment of skin sensitization potential [58,59]. In conclusion, THP-1 cells bring along various advantages for differentiation into dendritic cell surrogates.

In most studies to date, THP-1 cells have been differentiated into macrophages [60,61], and only very few publications have converted THP-1 cells into iDCs and mDCs (Table 1). Similar to human PBMCs, THP-1 cells could be differentiated into iDCs with cytokines, such as GM-CSF and IL-4 [62,63,64,65]. Maturation of THP-1 into mDCs was achieved via GM-CSF, IL-4, TNF-α, and ionomycin exposure in (serum-free) medium for 24 h to 72 h [62,66], or by cultivating THP-1-derived iDCs for an additional 24–72 h in (serum-free) medium supplemented with GM-CSF and/or IL-4, TNF-α, and ionomycin [64,65]. However, as these protocols differ in cell numbers, basic media composition, media supplementation, the number of days for differentiation, the frequency of media exchange and, most importantly, the cytokine concentrations in the differentiation cocktail, it becomes highly challenging to identify the appropriate method.

Thus, our aim was to establish robust and highly reproducible standard operating procedures addressing THP-1-derived iDC and mDC surrogates for in vitro toxicological studies as well as for investigating the underlying mechanisms of human skin sensitization and inflammation.

## 2. Results

The differentiation of THP-1 cells into dendritic cells has been described using RPMI or DMEM as a cultivation medium (Table 1). Both RPMI and DMEM are basal mediums which do not contain proteins or growth promoting agents and, as such, require supplementation with a serum, such as FBS. However, RPMI contains high concentrations of vitamins, as well as amino acids, such as asparagine, proline, biotin, and vitamin B12, which are not incorporated in DMEM [67]. On the other hand, DMEM contains higher concentrations of calcium (1.8 mM) and a lower concentration of phosphate (1 mM), compared to RPMI (0.8 mM calcium, 5 mM phosphate, respectively). While DMEM is selected for adherent cells, RPMI is widely used for suspension cells [68,69,70]. In line with this, RPMI is the recommended culture medium by the American Type Culture Collection (ATCC) and Deutsche Sammlung von Mikroorganismen und Zellkulturen (DSMZ) for THP-1 cells. Furthermore, PBMCs cultured in RPMI showed more efficient differentiation into DCs compared to PBMCs cultured in DMEM [71]. Based on these specifications, we decided to implement our differentiation protocols using RPMI.

The differentiation of THP-1 into iDCs and mDCs has been described using cytokine concentrations in ng/mL or in U/mL concentrations. However, by converting the indicated cytokine concentrations from ng/mL into U/mL and vice versa for the cytokines to be supplemented (Table 2), apparently significant differences become obvious. Thus, to systematically compare supplementation differences, we differentiated THP-1 cells into iDCs and mDCs by supplementing the medium with cytokines in either ng/mL or U/mL concentrations in parallel.

Differentiating THP-1 cells into iDCs in the presence of 100 ng/mL GM-CSF (≙900 U/mL) and 100 ng/mL IL-4 (≙2300 U/mL) resulted in a significantly higher number of cells expressing the surface markers CD54 (~95%), CD86 (~61%), and CD11b (~20%) (Figure 1A) compared to the undifferentiated control (CD54, ~58%; CD86, ~30%; CD11b, 3%). Furthermore, a pronounced higher geometric mean fluorescence intensity (GMFI) for HLA-DR (3.7-fold), CD54 (16.5-fold), and CD11b (3.1-fold) (Figure 1B) compared to the undifferentiated control could be detected. Supplementing the medium with 1500 U/mL GM-CSF and 1500 U/mL IL-4 induced the expression of CD54 (~88%), CD86 (~50%), and CD11b (~14%) on a significantly higher number of cells compared to the undifferentiated control (CD54, ~46%; CD86, ~24%; CD11b, ~3%) (Figure 1C) and a demonstrably higher GMFI for HLA-DR (4.1-fold), CD54 (14.6-fold), and CD11b (7.1-fold) (Figure 1D) compared to the undifferentiated control.

For the differentiation of THP-1 cells into mDCs, various protocols have been published. The main difference between those protocols is the direct differentiation of THP-1 cells into mDCs versus the differentiation of THP-1 cells into iDCs followed by further subsequent differentiation steps towards mDCs. Again, protocols using cytokine concentrations in ng/mL, as well as protocols based on concentrations in U/mL and various differentiation durations, have been published (Table 1).

Thus, we differentiated THP-1 in a one-step protocol into mDCs for 48 h with supplementation of 100 ng/mL GM-CSF (≙900 U/mL), 200 ng/mL IL-4 (≙4600 U/mL), 20 ng/mL (≙400 U/mL) TNF-α, and 200 ng/mL ionomycin and with supplementation of 1500 U/mL GM-CSF, 3000 U/mL IL-4, 2000 U/mL TNF-α, and 200 ng/mL ionomycin for 48 h as well as for 72 h. The surface marker expression of mDCs generated from THP-1 cells cultivated in serum-free medium for 48 h was significantly higher for CD54 (ng/mL, ~100%; U/mL, ~99%), CD86 (ng/mL, ~73%; U/mL, ~73%), CD11b (ng/mL, ~44%; U/mL, ~52%), and CD83 (ng/mL, ~29%; U/mL, ~50%), but significantly lower for CXCR4 (ng/mL, ~2%; U/mL, ~3%) compared to the undifferentiated controls (CD54, ~75–77%; CD86, ~44–48%; CD11b, ~3–4%; CD83, 0%; CXCR4, 33–35%) (Figure 2A,C). The differentiation of THP-1 cells into mDCs for 72 h using U/mL concentrations led to significantly higher numbers of cells expressing CD86 (~78%), CD11b (~49%), and CD83 (~35%), as well as CXCR4 (~70%), compared to the undifferentiated control (CD86, ~49%; CD11b, ~6%; CD83, ~0%; CXCR4, ~35%) (Figure 2E). Notably, a higher GMFI could only be observed for CD54 at a similar level (~ 415-fold) for all three differentiation approaches (Figure 2B,D,F).

Since the morphology of mDCs could be differentiated between floating and adherent populations (Figure 3D), we also investigated the impact of a floating versus an adherent status on surface marker expression (Figure 4). Both floating and adherent mDCs expressed the surface markers CD54 (floating, ~99; adherent, ~98%), CD86 (floating, ~81; adherent, ~80%), and CD83 (floating, ~25; adherent, ~26%) at significantly higher rates than undifferentiated THP-1 cells (CD54, ~79%; CD86, ~41%; CD83, ~0%). Furthermore, although the numbers of cells expressing CXCR4 and CD11b was significantly higher on floating mDCs (CXCR4, ~46%; CD11b, ~29%), compared to undifferentiated THP-1 cells (CXCR4. ~20%; CD11b, ~3%), the surface marker expression of CXCR4 (~29%) and CD11b (~3%) on the adherent mDC population was not significantly higher compared to undifferentiated THP-1 cells (CXCR4, ~20%; CD11b, ~3%) (Figure 4A). However, significant changes in the GMFI for CD54 were detected for floating (273-fold) as well as for adherent (225-fold) mDCs. Although the number of cells expressing HLA-DR was not increased, the GMFI for floating cells was elevated (2.3-fold) and was even higher for adherent mDCs (5.8-fold) compared to the undifferentiated control (Figure 4B). Thus, the differentiation of THP-1 cells with GM-CSF, IL-4, TNF-α, and ionomycin leads to two different populations, namely floating and adherent, which both reflect the marker expression of mature dendritic cells (CD54, CD86, and CD83), but display significant differences in CD11b and CXCR4 expression, which might be accompanied by different phagocytotic and migratory potential.

Next, we differentiated THP-1-derived iDCs further into mDCs (Figure 5). The differentiation of iDCs into mDCs led to a significantly higher number of cells expressing CD54 (~99%), CD86 (~54%), and CD11b (~21%) compared to undifferentiated THP-1 cells (CD54, ~63%; CD86, ~33%; CD11b, ~2%). Notably, the CXCR4 expression was completely diminished. The number of cells expressing CD11b was significantly higher in iDCs (~27%) as well as in mDCs (~21%) compared to undifferentiated THP-1 cells (~2%), but lower in mDCs (~21%) compared to iDCs (~27%). Expression of CD83 on mDCs from THP-1-derived iDCs could not be induced (Figure 5A). The GMFI for CD54 was 29.8-fold induced on iDCs and 128-fold induced on iDC-derived mDCs. However, the GMFI for HLA-DR was increased at similar levels for iDCs (8.3-fold) and mDCs (8.0-fold), and the GMFI induction for CD86 was lower for mDCs (16.4-fold), than for iDCs (26.2-fold) (Figure 5B). Furthermore, the morphology of mDCs differentiated from THP-1-derived iDCs, as depicted in Figure 3F, reveals mainly loosely adherent cell clusters.

To investigate the ability of THP-1-derived iDCs to identify potential sensitizers, iDCs in comparison to undifferentiated THP-1 cells were treated for 24 h with either 20 µM 1-chloro-2,4-dinitrobenzene (DNCB) or 380 µM nickel sulfate (NiSO_4_), the defined positive controls of the h-CLAT assay. As expected, the treatment of THP-1 cells with NiSO_4_ led to a significantly higher expression of the h-CLAT key markers CD54 (~98%) and CD86 (~55%) and induced the expression of the maturation marker CD83 (~17%). Furthermore, treatment of iDCs with NiSO_4_ resulted in a significant upregulation in CD54 (~97%), CD86 (~64%), and CXCR4 (~23%) (Figure 6A). The GMFI for CD54 was significantly higher (13.7-fold) after NiSO_4_ treatment on THP-1 cells, but not as high as on iDCs after NiSO_4_ treatment (29.2-fold) compared to untreated THP-1 cells. The GMFI for HLA-DR on iDCs was decreased after NiSO_4_ treatment (1.5-fold), but not as much as on THP-1 cells with (6.4-fold) or without NiSO_4_ treatment (5.6-fold) (Figure 6B). Furthermore, an increased GMFI (1.3-fold) for CD83 was observed after treatment of iDCs with NiSO_4_.

Treatment of THP-1 cells with 20 µM DNCB also resulted in significantly more cells expressing the h-CLAT markers CD54 (~98%) and CD86 (~96%), but significantly fewer cells expressing CXCR4 (~30%). Treatment of iDCs with DNCB resulted in significantly more cells expressing CD86 (~90%) (Figure 6C). Similar to the expression pattern of NiSO_4_-treated iDCs, treatment with DNCB led to a significantly higher GMFI for CD54 (3-fold) and lower GMFI for HLA-DR (1.9-fold) compared to untreated iDCs. However, compared to DNCB-treated THP-1 cells, the GMFI for HLA-DR is 3.3-fold higher, and the GMFI for CD11b is 7.2-fold higher on DNCB-treated iDCs (Figure 6D).

In order to prove the capability of our THP-1-derived DCs to phagocytose exogenous pathogen-derived particles, the DC surrogates were incubated with pHrodo Red-labeled zymosan, an insoluble β-1,3-glucan polysaccharide extracted from the cell wall of *Saccharomyces cerevisiae*. As pHrodo Red is a pH indicator dye, it is weakly fluorescent at neutral pH, but increases its fluorescence with decreasing pH during phagosomal acidification. As depicted in Figure 7 and as expected, significantly higher numbers of iDCs are able to phagocytose zymosan (~45%) compared to undifferentiated THP-1 cells (~17%) and to mDCs (~9%).

Moreover, the ability of iDCs to initiate T cell activation was investigated by analyzing the mRNA expression of the p40 chain of interleukin (IL)-12 via quantitative real-time PCR. For this, iDCs as well as undifferentiated THP-1 cells, were treated for 6 h with either 20 µM DNCB or 380 µM NiSO_4_, respectively. As expected, treatment of undifferentiated THP-1 cells with DNCB or NiSO_4_ did not significantly alter the expression of IL-12p40 (Figure 8A). In contrast, mRNA expression of IL-12p40 was significantly higher in iDCs after DNCB treatment (4.5-fold), and 1.3-fold higher after NiSO_4_ treatment, compared to the solvent control (Figure 8B), proving that differentiation of THP-1 cells into iDCs is required to study dendritic cell-mediated T cell activation.

## 3. Discussion

The aim of this study was to generate robust, highly reproducible, and cost-effective protocols providing THP-1-derived iDC as well as mDC surrogates for in vitro human-based toxicological studies and for investigating the underlying mechanisms of sensitization and inflammation. In mediating the immune response, dendritic cells undergo various phenotypical changes, such as the upregulation of co-stimulatory molecules, maturation markers, and receptors regulating migratory behavior. In order to evaluate a conclusive differentiation of THP-1 cells into iDCs and mDCs as well as their potential for toxicological studies, we focused mainly on the DC activation markers CD54 and CD86, CD11b as marker for phagocytosis, the maturation marker CD83, and the migration maker CXCR4.

The differentiation of THP-1 cells into iDCs resulted in a significant upregulation in the surface markers CD54, CD86, and CD11b. The upregulation of CD54 together with CD86 are the key readout parameters of the h-CLAT aiming to mimic dendritic cell activation in order to predict skin sensitization [58,59]. Furthermore, co-stimulatory molecules, such as CD86, as well as CD80, often working in tandem, are upregulated during DC maturation, promoting CD 4^+^ T cell activation [34,72]. The upregulation of CD86 as well as CD80 have been shown for THP-1-derived iDCs [62,64]. However, contrarily, Galbiati et al. described CD80 as well as CD86 expression below 15% on iDCs [65], not matching our results. It is known that CD11b plays an important role in phagocytosis [73], and its upregulation on THP-1-derived iDCs has been revealed by Czernek et al. [64], a study which confirms our data. Furthermore, HLA-DR is one of the MHC class II cell surface receptors, presenting the internalized and processed antigens to CD 4^+^ T cells [74]. Even though our results displayed a low number of iDCs expressing HLA-DR, the GMFI for HLA-DR was significantly higher compared to undifferentiated cells, verifying an upregulation of HLA-DR molecules on HLA-DR-positive iDCs. This result matches the findings of Czernek et al., reporting a substantially higher MFI for HLA-DR on iDCs; unfortunately, Czernek et al. did not indicate the number of positive cells for HLA-DR [64]. However, as long as iDCs have not been exposed to antigens, MHC II molecules are retained by the invariant chain li in the late endosomal compartment [14]. In addition, it has been reported that only very few MHC II molecules are localized on the membrane of iDCs, and up to 75% of all MHC II molecules reside in the antigen processing compartments [15], confirming our results revealing low numbers of iDCs expressing HLA-DR on their surface. Furthermore, the differentiation of THP-1 cells into iDCs did not induce the expression of CD83, a principal marker for cell maturation [75], matching the literature [62] and proving their immature status. To date, most protocols differentiating THP-1 cells into iDCs were performed in T75 flasks and 20 mL medium [62,65,66]. In contrast, we are the first to prove the differentiation of THP-1 cells into iDCs in T25 flasks, using 5 mL medium and, thus, only one quarter of the amount of the required cytokines, confirming our cost-effective approach.

For the generation of THP-1-derived mDCs, various protocols have been published, differentiating THP-1 cells directly into mDCs for 24 h [62], 48 h [62,66], 72 h [62], or generating mDCs from iDCs [63,64,65]. Differentiating THP-1 cells directly into mDCs for 48 h resulted in significantly higher numbers of cells expressing CD54, CD86, CD11b, and CD83, but a significantly lower (almost none) expression of CXCR4. Contrarily, the differentiation of THP-1 cells into mDCs for 72 h led to significantly enhanced numbers of CXCR4 expressing cells compared to the undifferentiated control. Furthermore, CXCR4 is one receptor for CXCL12/SDF-1, which is constitutively expressed and secreted in lymphoid tissues and other non-lymphoid tissues by bone marrow-, lymph node-, skin-, muscle-, and lung-derived fibroblasts, as well as by endothelial cells, liver and kidney cells, and the central nervous system [76,77,78,79]. Furthermore, various organs respond to tissue damage by increasing SDF-1 expression and secretion via hypoxia-inducible factor-1 (HIF-1) binding to the hypoxia-responsive region of the SDF-1 promotor [80]. Based on knockout studies or pharmacological blockade, CXCR4 has been proven to have a key role in DC differentiation [81], as well as in DC and Langerhans cell migration [23,24,82]. We are the first to prove CXCR4 expression on THP-1-derived mDCs. Coherent, upregulation of CXCR4 surface expression on mature dendritic cells has been shown for PBMC-derived dendritic cells [83] as well as for bone marrow-derived dendritic cells [84]. Furthermore, low CXCR4 mRNA levels for PBMC-derived iDCs and high CXCR4 levels for PBMC-derived mDCs have been detected by Sallusto et al. [21], confirming the CXCR4 expression pattern on our iDCs and mDCs. However, CXCR4 is not only expressed on mDCs, but also on naïve T cells and B lymphocytes [85], which may favor co-localization of those cells at sites where SDF-1 is secreted due to inflammation and tissue damage. Noteworthily, the direct differentiation of THP-1 cells into mDCs for 72 h led to two different subpopulations, namely floating and adherent cells. In both cell populations, CD83, the marker molecule for mature dendritic cells, was expressed by a similar quantity of cells, indicating no difference in maturation status. While the floating cells showed significantly higher expression of CXCR4 and CD11b than the undifferentiated THP-1 cells, the expression of CXCR4 and CD11b on adherent mDCs was only marginally higher than on undifferentiated THP-1 cells and significantly lower compared to the floating mDCs. Since CXCR4 is necessary for migration [23,24,82] and CD11b is involved in phagocytosis [73], it is coherent with the morphology and higher expression on floating cells which might be still in a steady state for the phagocytosis of antigens and migration to distinct sides. Furthermore, mDCs generated from THP-1-derived iDCs also display a different morphology compared to the mDCs which have been generated directly from THP-1 cells. While the directly generated mDCs were mostly strongly adherent and developed dendritic shaped branches, mDCs generated from iDCs formed loosely adherent cell clusters. Furthermore, CD11b expression was slightly lower than on iDCs and they expressed neither CD83 nor CXCR4. Based on these results we are confident that mDCs generated from THP-1-derived iDCs are not mDCs. Unfortunately, all publications which generated mDCs from THP-1-derived iDCs did not investigate the expression of CD83 or CXCR4, as successful maturation was assumed from high CD80 and CD86 expression [63,64,65]. Thus, only mDCs generated directly from THP-1 cells match the dendritic morphology and express the relevant maturation markers CD86 and CD83 as well as the migration marker CXCR4. However, for CXCR4 expression, it is mandatory to differentiate the THP-1 cells for 72 h and not as stated otherwise for 24 h or 48 h. Furthermore, comparing the differentiation results for iDCs and mDCs, expression levels for CD54, CD11b, and CD83 were different using cytokine concentrations indicated in ng/mL versus U/mL. Since the biological activity differs from supplier to supplier and occasionally between lots, the indication of applied cytokine concentrations in U/mL is more precise and strongly advised for reproducible data. Complementary to the iDC protocols, most protocols for mDC generation were performed in T75 flasks in a volume of 20 mL medium. We are the first to prove differentiation of THP-1 cells into iDCs in T25 flasks, using 5 mL medium and, thus, only one quarter of the cytokines, thereby generating a cost-effective protocol.

As mentioned before, the upregulation of co-stimulatory molecules, as well as maturation markers in response to sensitizing and inflammation inducing agents, has become one of the key parameters to identify and predict the potential sensitizing and inflammatory capacity of substances. One of the most prominent assays is the h-CLAT, predicting sensitizers via CD54 and CD86 upregulation. The accuracy of the h-CLAT to distinguish sensitizers from non-sensitizers has been calculated as being between 76% and 83% [58,86,87]. In order to be classified as sensitizer, the relative fluorescence intensity (RFI) has to exceed a defined threshold, namely CD54 > 200 or CD86 > 150 [56,59]. However, detection of chemicals as false-negatives in the h-CLAT have been reported [54,87]. Our data reveals that the expression of CD54 as well as CD86, including percent positive cells and GMFI, is significantly higher for iDCs after sensitizer treatment compared to undifferentiated cells, which might allow a higher accuracy in detecting and subsequently categorizing sensitizers by decreasing rates of false-negative results and which might provide benefits for animal-free toxicological studies.

In order to demonstrate the functionality of our iDCs, their capability to phagocytose exogenous pathogen-derived particles as well as their potential to activate T cells was investigated. In their immature state, DCs are able to endocytose pathogens, which are further processed, initiating DC activation and maturation accompanied by the expression of surface markers, such as MHC II, CD54, and CD86. During the maturation process this ability decreases as DCs acquire primarily potent antigen presenting functions [12,88]. To assess the phagocytic potential of our DCs, cells were treated with zymosan, derived from the cell wall of *Saccharomyces cerevisiae*. While some undifferentiated THP-1 cells were able to phagocyte zymosan (17%), the number of phagocytotic iDCs was significantly higher (45%), whereas the number of mDCs phagocyting zymosan decreased (9%). In line with our findings, high phagocytotic capability for iDCs has been reported for PBMC-derived as well as bone-marrow-derived iDCs [89,90,91]. Furthermore, lower phagocytotic capability of mDCs upon maturation has been shown for mDCs derived from PBMCs [91].

In order to activate naïve CD4^+^ T cells, DCs secrete IL-12, leading to upregulation of the transcription factor T-box expressed in T cells (T-bet) promoting their differentiation into interferon-γ producing T helper 1 cells (Th1) [92,93]. Treatment of our THP-1-derived iDCs with DNCB resulted in significantly higher mRNA levels of IL-12p40 and in moderate higher mRNA levels of IL-12p40 in the presence of NiSO_4_. Overall, we are able to demonstrate and prove the potential of THP-1-derived iDCs to induce T cell activation. Previous studies have shown IL-12p40 expression induction upon NiSO_4_ treatment of PBMC-derived iDCs [94,95], but not after DNCB treatment [94,96]. Contrarily, cutaneous treatment of mice with DNCB led to significantly enhanced IL-12p40 mRNA levels in local lymph nodes [97,98], as well as in spleens and the skin [99], tending to confirm our findings.

In conclusion, we were able to downscale the differentiation approaches by three-quarters, thereby generating robust, highly reproducible, and cost-effective protocols providing THP-1-derived iDC and mDC surrogates. The strong induction of CD54 as well as CD86 expression on iDCs after sensitizer treatment are applicable for in vitro toxicological studies, identifying potential sensitizing or inflammatory compounds and, in further steps, for assessing the anti-inflammatory potential of novel drug candidates. Based on the observed expression induction rates of CD11b, CD83, and CXCR4 as well as the IL-12p40 expression upon sensitizer treatment and the phagocytotic capability, our iDC and mDC surrogates are beneficial to study the molecular mechanisms of dendritic cell-mediated phagocytosis [73,100], dendritic cell maturation [101], as well as migration [22] and, furthermore, their use as therapeutic model systems in various disorders, such as sensitization, inflammation [102,103], as well as cancer [104] and the tumor microenvironment [105], should not be discounted.

## 4. Materials and Methods

### 4.1. Cell Line Cultivation

The human monocytic leukemia cell line THP-1 (#TIB202, LOT:70025047) was purchased from ATCC (Manassas, VA, USA) (The THP-1 cells were maintained in T75 flasks (Greiner, #658195, Frickenhausen, Germany) in 20 mL RPMI (Gibco, #22400089, Grand Island, NY, USA) supplemented with 10% FBS (Gibco, #10270-106), 1% penicillin–streptomycin (PenStrep) (Gibco, #15140122), and 0.05 mM 2-mercaptoethanol (Gibco, #21985023) in a humidified incubator at 37 °C and 5% CO_2_. Cell density was maintained between 1 × 10^5^ cells/mL and 5 × 10^5^ cells/mL and cells were split every 2–3 days.

### 4.2. Differentiation of THP-1 Cells into iDCs

For the generation of iDCs 2 × 10^5^ THP-1 cells/mL were seeded in 5 mL RPMI supplemented with 10% FBS, 1% PenStrep, and 0.05 mM 2-mercaptoethanol into a T25 flask. For differentiation, according to the published ng/mL concentrations, the following concentrations of cytokines were added: 100 ng/mL (=900 U/mL) rhGM-CSF (ImmunoTools, #11343125, Friesoythe, Germany), and 100 ng/mL (=2300 U/mL) rhIL-4 (ImmunoTools, #11340045). For differentiation using U/mL concentrations, 1500 IU/mL rhGM-CSF (ImmunoTools, #11343125) and 1500 IU/mL rhIL-4 (ImmunoTools, #11340045) were added. The cells were incubated for 5 days at 37 °C, 5% CO_2_, with medium exchange and addition of fresh cytokines on day 3.

### 4.3. Differentiation of THP-1 Cells into mDCs

For differentiation of THP-1 cells into mDCs 2 × 10^5^ cells/mL in 5 mL or 20 mL serum-free RPMI supplemented with 1% PenStrep and 0.05 mM 2-mercaptoethanol was placed into a T25 flask or T75 flask. For differentiation according to ng/mL cytokine concentrations, the following concentrations were added: 100 ng/mL (=900 U/mL) rhGM-CSF (ImmunoTools, #11343125), 200 ng/mL (=4600 U/mL) rhIL-4 (ImmunoTools, #11340045), 20 ng/mL (=400 U/mL) TNF-a (PromoKine, #C63719), and 200 ng/mL ionomycin (Sigma-Aldrich, #I0634). For differentiation using U/mL concentrations, the following cytokines were added: 1500 IU/mL rhGM-CSF (ImmunoTools, #11343125), 3000 IU/mL rhIL-4 (ImmunoTools, #11340045), 2000 IU/mL TNF-α (PromoKine, #C63719, Heidelberg, Germany), and 200 ng/mL ionomycin (Sigma-Aldrich, #I0634, Darmstadt, Germany). The cells were cultivated for 48 h and 72 h at 37 °C, 5% CO_2_. For flow cytometry analysis, adherent cells were detached with accutase (Sigma-Aldrich, #A6964).

### 4.4. Differentiation of THP-1-Derived iDCs into mDCs

The THP-1-derived iDCs were generated as stated in Section 4.2. On day 5, for further differentiation into mDCs, the medium was removed, and fresh medium containing 100 ng/mL (=900 U/mL) rhGM-CSF (ImmunoTools, #11343125), 200 ng/mL (=4600 U/mL) rhIL-4 (Immuno-Tools, #11340045), 20 ng/mL (=400 U/mL) TNF-α (PromoKine, #C63719), and 200 ng/mL ionomycin (Sigma-Aldrich, #I0634) was added. The cells were cultivated for 48 h at 37 °C, 5% CO_2_. For flow cytometry analysis, adherent cells were detached with accutase (Sigma-Aldrich, #A6964).

### 4.5. Sensitization Assay According to the h-CLAT

The THP-1 cells or THP-1-derived iDCs (see Section 4.2) were seeded and treated accordingly to the h-CLAT assay. Briefly, 1 × 10^6^ cells/mL were seeded in 1 mL RPMI supplemented with 10% FBS, 1% PenStrep, and 0.05 mM 2-mercaptoethanol into a 24-well plate. Cells were treated with 20 µM 1-chloro-2,4-dinitrobenzene (DNCB) (Sigma-Aldrich, #237329, Darmstadt, Germany) and 380 µM nickel sulfate (NiSO_4_) (Sigma-Aldrich, #227676) or their respective solvent control, namely dimethylsulfoxide (DMSO) or Dulbecco’s phosphate buffered saline (PBS). After 24 h, the cells were harvested, and surface marker expression was determined via flow cytometry (see Section 4.6).

### 4.6. Surface Marker Detection via Flow Cytometry

Cells were harvested after differentiation and washed thrice in autoMACS running buffer (Miltenyi Biotec, #130-091-221, Gladbach, Germany). Cells were transferred to 96-well plates with 2 × 10^5^ cells for each antibody panel. Cells were stained with the following antibodies: diluted 1:50, REA Control (S)-VioGreen (Miltenyi Biotec, #130-113-444), REA Control (S)-PE (Miltenyi Biotec, #130-113-438), REA Control (S)-APC (Miltenyi Biotec, #130-113-434); REA Control (S)-PE-Vio770, (Miltenyi Biotec, #130-113-440); HLA-DR-VioGreen (Miltenyi Biotec, #130-111-795), CD54-APC (Miltenyi Biotec, #130-121-342); CXCR4-PE-Vio770 (Miltenyi Biotec, #130-116-161); CD11b-VioGreen (Miltenyi Biotec, #130-110-617); CD83-PE (Miltenyi Biotec, #130-110-561); CD86-APC (Miltenyi Biotec, #130-116-161) for 10 min at 4 °C in the dark. Afterwards, cells were washed thrice with autoMACS running buffer and stained with DAPI (Sigma, #D9542), to exclude dead cells for the determination of the cell viability.

### 4.7. Phagocytosis Assay

Phagocytosis assays were performed using pHrodo red zymosan particles (Invitrogen^TM^, #P35364, Waltham, MA, USA), dissolved in RPMI supplemented with 10% FBS, 1% PenStrep, and 0.05 mM 2 mercaptoethanol at a concentration of 0.5 mg/mL. Cells were harvested after differentiation and 2 × 10^5^ cells were resuspended in 100 µL pHrodo red zymosan particles and seeded into 96-well plates. The cells were cultivated for 1 h at 37 °C, 5% CO_2_. Then, DAPI was added before the analysis of 10,000 living cells per sample via flow cytometry (Ex/Em, 560/585).

### 4.8. Analysis of IL-12p40 mRNA Expression by Quantitative Real-Time PCR

The THP-1 cells or THP-1-derived iDCs (see Section 4.2) were seeded and treated according to the h-CLAT assay. Briefly, 1 × 10^6^ cells/mL were seeded in 1 mL RPMI supplemented with 10% FBS, 1% PenStrep, and 0.05 mM 2-mercaptoethanol into a 24-well plate. Cells were treated with 20 µM 1-chloro-2,4-dinitrobenzene (DNCB) (Sigma-Aldrich, #237329) and 380 µM nickel sulfate (NiSO4) (Sigma-Aldrich, #227676) or their respective solvent control, namely dimethylsulfoxide (DMSO) or Dulbecco’s phosphate buffered saline (PBS) for 6 h. Total RNA was extracted according to the manufacturer’s instructions using the RNeasy MiniKit (Qiagen, #74104, Hilden, Germany). The RNA concentration was determined by OD260/280 measurement using the Tecan Spark NanoQuant Plate. A total of 1 µg of RNA was reverse transcribed using the QuatiTect Reverse Transcription Kit (Qiagen, #205311). Quantitative real-time PCR (qPCR) reactions were performed for 50 ng cDNA in triplicate for each sample on a qTower^3^ G (Analytikjena, Jena, Germany), using Luna Universal qPCR Master Mix (NEB, #M3003L, Ipswich, MA, USA). The specific primers used were GAPDH (forward, 5′-TGCACCACCAACTGCTTAGC-3′; reverse, 5′-GGCATGGACTGTGGTCATGAG-3′) and IL-12p40 (forward, 5′-TGTCGTAGAATTGGATTGGTATC-3′; reverse, 5′-AACCT GCCTCCTTTGTG-3′). After amplification, a threshold was set for each gene and Ct values were calculated for all samples.

### 4.9. Statistical Analysis

Statistical analysis was performed using GraphPad Prism version 8.4.3 (GraphPad Software, Inc., San Diego, CA, USA). Statistical significance was determined using two-way ANOVA, Sidak’s multiple comparisons test or Tukey’s multiple comparisons test. Significance was defined as * = *p* ≤ 0.05; ** = *p* ≤ 0.01; *** = *p* ≤ 0.001; **** = *p* ≤ 0.0001.

## Figures and Tables

**Figure 1 ijms-24-01452-f001:**
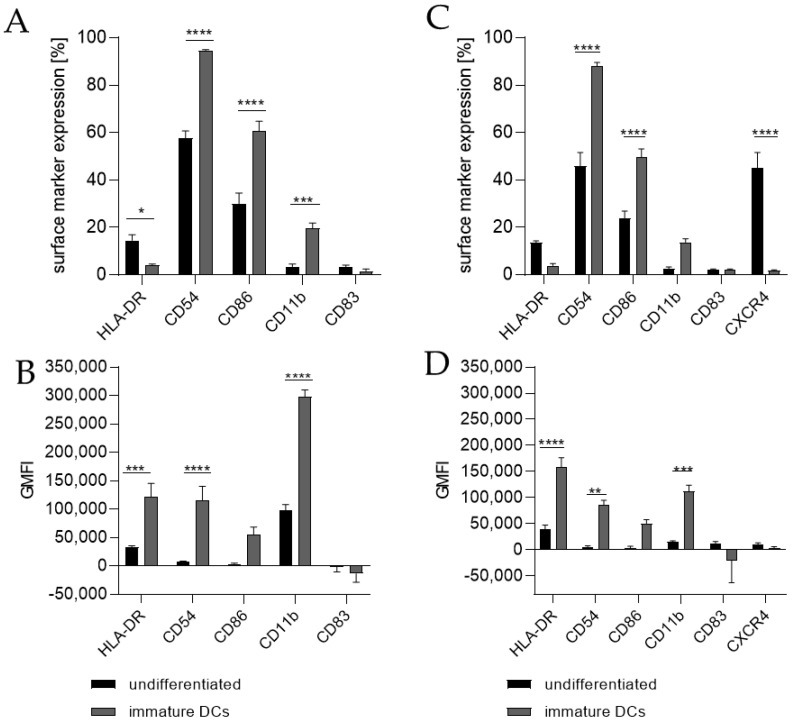
Surface marker expression of THP-1-derived iDCs, as follows: ng/mL (**A**,**B**) versus U/mL (**C**,**D**). Here, 2 × 10^5^ THP-1 cells/mL were seeded in 5 mL RPMI supplemented with 10% FBS, 1% PenStrep, and 0.05 mM 2-mercaptoethanol into a T25 flask. (**A**,**B**) For differentiation into iDCs, 100 ng/mL rhGM-CSF and 100 ng/mL rhIL-4 were added. (**C**,**D**) For differentiation into iDCs, 1500 U/mL rhGM-CSF and 1500 U/mL rhIL-4 were added. Cells were cultivated for 5 days, with medium exchange and addition of fresh cytokines after 72 h. Surface marker expression of at least 10,000 viable cells was analyzed via flow cytometry. Error bars indicate the standard errors of the mean (n = 3 independent experiments with * = *p* ≤ 0.05, ** = *p* ≤ 0.01, *** = *p* ≤ 0.001, and **** = *p* ≤ 0.0001).

**Figure 2 ijms-24-01452-f002:**
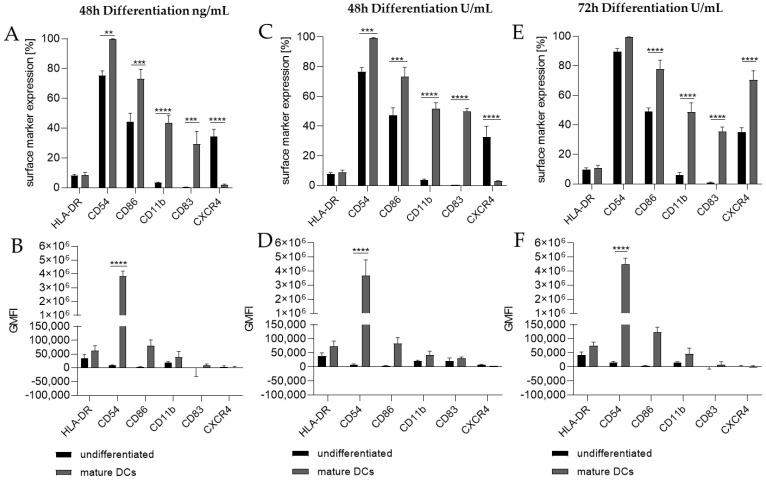
Surface marker expression of THP-1-derived mDCs, as follows: ng/mL (**A**,**B**) versus U/mL (**C**–**F**). Here, 2 × 10^5^ THP-1 cells/mL were seeded in 5 mL serum-free RPMI supplemented with 1% PenStrep, and 0.05 mM 2-mercaptoethanol into a T25 flask. (**A**,**B**) For differentiation into mDCs, 100 ng/mL rhGM-CSF, 200 ng/mL rhIL-4, 20 ng/mL TNF-α, and 200 ng/mL ionomycin were added. (**C**,**D**) For differentiation into mDCs, 1500 U/mL rhGM-CSF, 3000 U/mL rhIL-4, 2000 U/mL TNF-α, and 100 ng/mL ionomycin were added. Cells were cultivated for 48 h or 72 h. Surface marker expression of at least 10,000 viable cells was analyzed via flow cytometry. Error bars indicate the standard errors of the mean (n = 3 independent experiments, with ** = *p* ≤ 0.01, *** = *p* ≤ 0.001, and **** = *p* ≤ 0.0001).

**Figure 3 ijms-24-01452-f003:**
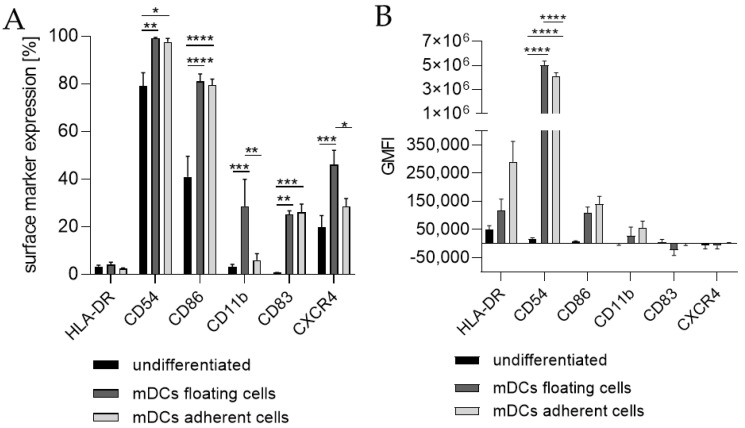
Surface marker expression of THP-1-derived mDCs. Floaters versus adherent cells. (**A**) Surface marker expression [%]. (**B**) Geometric mean fluorescence intensity (GMFI) 2 × 10^5^ THP-1 cells/mL were seeded in 20 mL serum-free RPMI supplemented with 1% PenStrep and 0.05 mM 2-mercaptoethanol into a T75 flask. For differentiation, 1500 U/mL rhGM-CSF, 3000 U/mL rhIL-4, 2000 U/mL TNF-α, and 200 ng/mL ionomycin were added. Cells were cultivated for 72 h at 37 °C, 5% CO_2_. Surface marker expression of at least 10,000 viable cells was analyzed via flow cytometry. Error bars indicate the standard errors of the mean (n = 3 independent experiments with * = *p* ≤ 0.05; ** = *p* ≤ 0.01; *** = *p* ≤ 0.001; and **** = *p* ≤ 0.0001).

**Figure 4 ijms-24-01452-f004:**
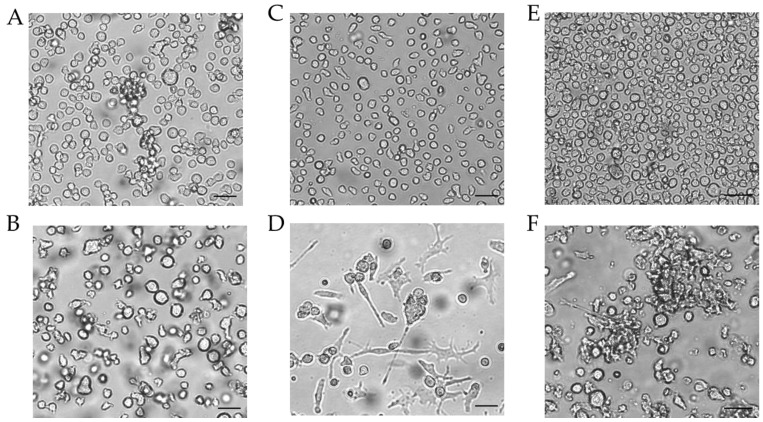
Morphology of THP-1-derived iDCs and mDCs using cytokine concentrations in U/mL. Morphology of undifferentiated THP-1 cells (**A**,**C**,**E**) according to the respective culture conditions of the differentiated cells. (**B**) Morphology of THP-1-derived immature dendritic cells (iDCs) (T25, U/mL, 5 d). (**D**) Morphology of THP-1-derived mDCs (T25, U/mL, 72 h). (**F**) Morphology of mDCs generated from THP-1-derived iDCs (T25, ng/mL, 48 h). Scale bar = 50 µm.

**Figure 5 ijms-24-01452-f005:**
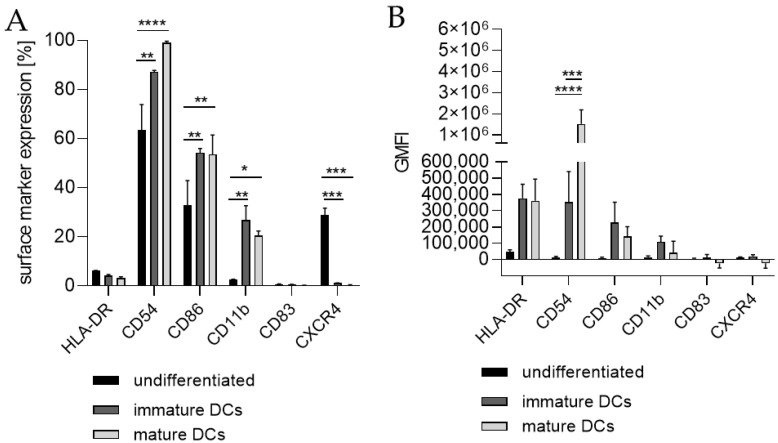
Surface marker expression of mDCs generated from-THP-1-derived iDCs. (**A**) Surface marker expression [%]. (**B**) Geometric mean fluorescence intensity (GMFI). Here, 2 × 10^5^ THP-1 cells/mL were seeded in 5 mL RPMI supplemented with 10% FBS, 1% PenStrep, and 0.05 mM 2-mercaptoethanol into a T25 flask and differentiated into iDCs (see Section 4.2). For further differentiation into mDCs, the medium was removed and fresh medium containing 100 ng/mL rhGM-CSF, 200 ng/mL rhIL-4, 20 ng/mL TNF-α, and 200 ng/mL ionomycin was added. Cells were cultivated for 48 h at 37 °C, 5% CO_2_. Surface marker expression of at least 10,000 viable cells was analyzed via flow cytometry. Error bars indicate the standard errors of the mean (n = 2 independent experiments with * = *p* ≤ 0.05, ** = *p* ≤ 0.01, *** = *p* ≤ 0.001, and **** = *p* ≤ 0.0001).

**Figure 6 ijms-24-01452-f006:**
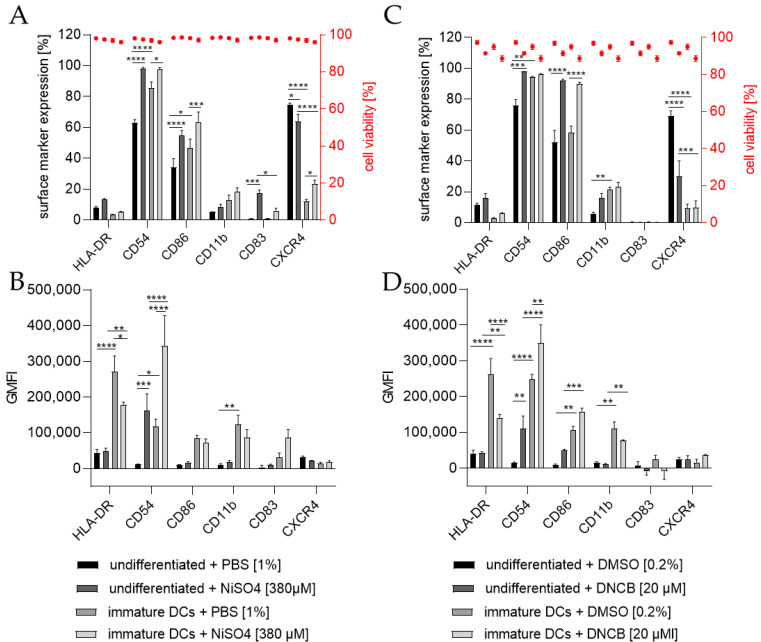
Surface marker expression of THP-1 cells or iDCs after sensitization according to the h-CLAT assay. (**A**,**C**) surface marker expression [%]. (**B**,**D**) Geometric mean fluorescence intensity (GMFI). Here, THP-1 cells or THP-1-derived iDCs (see Section 4.2) were seeded with 1 × 10^6^ cells/mL in 1 mL RPMI supplemented with 10% FBS, 1% PenStrep, and 0.05 mM 2-mercaptoethanol into a 24-well plate. Cells were treated with 20 µM 1-Chloro-2,4-dinitrobenzene (DNCB) and 380 µM nickel sulfate (NiSO_4_) or their respective solvent control, namely DMSO or PBS. Surface marker expression of at least 10,000 viable cells was analyzed via flow cytometry. Error bars indicate the standard errors of the mean (n = 3 independent experiments with * = *p* ≤ 0.05, ** = *p* ≤ 0.01, *** = *p* ≤ 0.001, and **** = *p* ≤ 0.0001).

**Figure 7 ijms-24-01452-f007:**
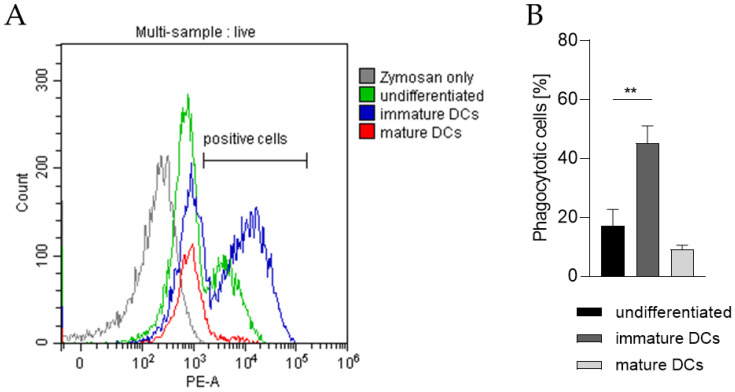
Phagocytotic capability of undifferentiated THP-1 cells, iDCs, and mDCs. Here, iDCs, as well as mDCs, were differentiated according to the U/mL protocols (see Section 4.2 and Section 4.3). After 5 days and 72 h respectively, 2 × 10^5^ cells were resuspended in 100 µL pHrodo Red zymosan bioparticles, seeded into 96-well plates, and cultivated for 1 h at 37 °C, 5% CO_2_. Then, DAPI was added before analysis of 10,000 living cells per sample via flow cytometry (Ex/Em, 560/585). (**A**) Gating strategy. (**B**) Number of cells (%) positive for phagocytosis. Error bars indicate the standard errors of the mean (n = 3 independent experiments, with ** = *p* ≤ 0.01)).

**Figure 8 ijms-24-01452-f008:**
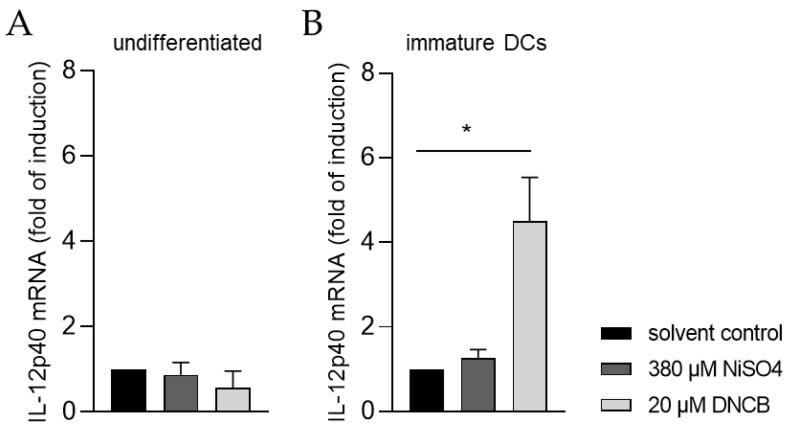
IL-12p40 mRNA expression of (**A**) THP-1 cells and (**B**) immature dendritic cells (iDCs) after sensitizer treatment. Here, THP-1 cells or THP-1-derived iDCs (see Section 4.2) were seeded with 1 × 10^6^ cells/mL in 1 mL RPMI supplemented with 10% FBS, 1% PenStrep, and 0.05 mM 2-mercaptoethanol into a 24-well plate. Cells were treated with 20 µM DNCB or 380 µM NiSO4 for 6 h. Results were expressed as fold of induction compared to the solvent control and normalized to the expression of the housekeeping gene GAPDH. Error bars indicate the standard errors of the mean (n = 3 independent experiments with * = *p* ≤ 0.05).

**Table 1 ijms-24-01452-t001:** Literature review of the differentiation of THP-1 into iDCs or mDCs.

Cultivation Conditions	iDCs	mDCs	Ref.
Cytokine concentrations	100 ng/mL (1500 U/mL) GM-CSF 100 ng/mL (1500 U/mL) IL-4	100 ng/mL (1500 U/mL) GM-CSF 200 ng/mL (3000 U/mL) IL-4 20 ng/mL (2000 U/mL) TNF-α 200 ng/mL ionomycin	[62]
Medium Cell number Time of differentiation	RPMI, 10% serum, 2 × 10^5^ cells/ ml, 20 mL 5 d of cultivation	RPMI, serum-free, 2 × 10^5^ cells/ ml, 20 mL 24–72 h of cultivation	
Cytokine concentrations	150 ng/mL GM-CSF 50 ng/mL IL-4	Exposure of generated iDCs to 10 ng/mL IL-1β 10 ng/mL TNF-α 2 µg/mL PGE_2_ 25% MCM Or 1 µg/mL LPS	[63]
Medium Cell number Time of differentiation	RPMI, 10% serum, 5 × 10^5^ cells/ ml 7 d of cultivation	RPMI, 10% serum Cell number not indicated 48 h of cultivation	
Cytokine concentrations	-	100 ng/mL (1500 U/mL) GM-CSF 200 ng/mL (3000 U/mL) IL-4 20 ng/mL (3000 U/mL) TNF-α 200 ng/mL ionomycin	[66]
Medium Cell number Time of differentiation	-	DMEM, serum-free 2 × 10^5^ cells/ ml, 20 mL 48 h of cultivation	
Cytokine concentrations	100 ng/mL GM-CSF 100 ng/mL IL-4	Exposure of generated iDCs to:100 ng/mL GM-CSF 100 ng/mL IL-4 20 ng/mL TNF-α 200 ng/mL ionomycin	[64]
Medium Cell number Time of differentiation	RPMI, 10% serum Not indicated 5 d of cultivation	RPMI, serum-free Not indicated 72 h of cultivation	
Cytokine concentrations	1500 U/mL GM-CSF 1500 U/mL IL-4	Exposure of generated iDCs to: 3000 U/mL IL-4 2000 U/mL TNF-α 200 ng/mL ionomycin	[65]
Medium Cell number Time of differentiation	RPMI, 10% serum 2 × 10^5^ cells/mL, 20 mL 5 d of cultivation	RPMI, 10% serum Not indicated 24 h of cultivation	

**Table 2 ijms-24-01452-t002:** Cytokine concentrations applied for the differentiation of THP-1 cells into iDCs or mDCs, calculated by referring to the biological activity published by the manufacturer.

Cytokine	ng/mL	U/mL
GM-CSF [ImmunoTools, #1343125]	100 ng/mL 166.67 ng/mL	900 U/mL 1500 U/mL
IL-4 [ImmunoTools, #11340045]	100 ng/mL 200 ng/mL 65.22 ng/mL 130.44 ng/mL	2300 U/mL 4600 U/mL 1500 U/mL 3000 U/mL
TNF-α [PromoKine, #C-63719]	20 ng/mL 100 ng/mL	400 U/mL 2000 U/mL

## Data Availability

Not applicable.

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
