# Peer review of "The Monocytic Cell Line THP-1 as a Validated and Robust Surrogate Model for Human Dendritic Cells"

_ijms, 2023, doi:10.3390/ijms24021452_

Round 1

Reviewer 1 Report

accept for publication

Author Response

The authors would like to thank for this positive feedback.

Reviewer 2 Report

Holken and Teusch describe a protocol for the differentiation of dendritic cells from the monocyte cell line THP1. As also acknowledged by the authors, this is not particularly innovative as there have been several studies were this has been done. However, the detailed characterization and direct comparison of protocols to generate immature and mature DCs (directly and via immature DCs) will be of value to the field. The paper is very well written and the data are solid. However, I think a few more experiments and controls will help to really make this a standard protocol for the differentiation of Thp1 into DCs:

-The main limitation of this study is that the phenotypical characterization is limited to flow cytometry of surface markers. Some more ‘functional’ experiments for endocytosis/phagocytosis (using fluorescently labelled dextran/OVA/beads) and T cell activation (using mixed lymphocyte reactions) are easy to perform and will make the characterization of the DCs a lot more thorough.

-Similarly, it would be important to determine the cytokine production of DC cytokines, especially IL12 (and IL10) given its role in T cell activation, by ELISA or rt-PCR.

-Table 2 shows differences in cytokine concentrations used in published protocols. Please add references to this. Also: how was the ng/ml concentration converted to U/ml. The conversion factors differ between manufacturers and batches, right? Was this accounted for?

-For all figures, it would be nice to show the flow cytometry gating strategy and representative histrograms. Was a live/death marker included? Are the GMFI calues calculated on all cells or only on positive cells?

-In figure 2: how mature are these mDCs relative to iDCs and other mDCs? It would be could to compare them for example with the iDCs of panel 1 or the mDCs of figure 5.

-Figure 3A shows surprisingly that integrin CD11b is lower in the adherent than in the floating cells. However: the protease accutase was used for cell detachment. Does this perhaps cleave integrins, thereby leading to lower surface expression? The authors should exclude such an artefact.

Minor points:

-The Introduction section is very elaborate, but sometimes superfluous and some details can be removed (for instance the molecular cascade of MHC class II trafficking). Moreover, the Introduction is focussed on MHC-II presentation of CD4 T cells, but what about MHC-I and CD8 T cells?

-Page 2, line 74: “restored” should be “stored”.

-The immunogen DNCB should be introduced.

Reviewer 3 Report

The manuscript [ijms-2072135], entitled “The monocytic cell line THP-1 as a validated and robust surrogate model for human dendritic cells” by Dr. Teusch et al., reports establishment of a robust and highly reproducible protocol for converting THP-1 cells into immature and mature dendritic cells, which would facilitate vital studies including  investigations on molecular mechanisms of dendritic cell maturation, phagocytosis, migration and therapeutic targets of cancer.

The study was well designed and appeared properly executed step by step. The figures and tables were well prepared and very informative in presentation of the data from this study. This reviewer is convinced that the findings of this study would be of wild interest for investigators working in this and other relevant fields.

Minor concerns are listed below for consideration in revision.

      Major concerns:

1.      None.

Minor concerns:

2.      Lines 183, 215, 243, 274, and 305: Suggested to remove the repeated statement “Statistical analysis was performed using Two-Way-ANOVA” but keep the Turkey’s test statement without the parentheses at the places.

3.      Insert a subsection of statistical analysis under the M&M, where more specifics may be offered on the description of the used two-way ANOVA model.

Author Response

The authors would like to thank for the valuable feedback.

In detail, the reviewer suggested the following:

  1. Lines 183, 215, 243, 274, and 305: Suggested to remove the repeated statement “Statistical analysis was performed using Two-Way-ANOVA” but keep the Turkey’s test statement without the parentheses at the places.

The authors have the removed this statement accordingly.

  1. Insert a subsection of statistical analysis under the M&M, where more specifics may be offered on the description of the used two-way ANOVA model.

A section in the method part has been added as suggested.

Round 2

Reviewer 2 Report

The authors have addressed my comments and I support pulication.

I only have a single minor comment concerning my point 7 regarding MHC-I presentation. Whereas I agree with the authors that all nucleated cells present endogenous antigens in MHC-I, the so-called cross-presentation of exogenous antigens in MHC-I is a key hallmark of DCs and necessary for activation of naive CTLs. This caveat should be acknowledged in the text somewhere.

Author Response

The authors would like to thank again for the constructive and competent feedback.

As suggested we have included the information on MHC-I briefly in the introduction including two references as highlighted in green in the current manuscript version.